# Gender-Specific Effect of Couple-Based Intervention on Behavioral and Psychological Outcomes of Older Adults with Type 2 Diabetes during the COVID-19 Partial Lockdown in Guangzhou, China

**DOI:** 10.3390/healthcare10112290

**Published:** 2022-11-15

**Authors:** Conghui Yang, Yingxin Xu, Jingyi Zhi, Huiqiong Zheng, Zhenhua Sun, Xueji Wu, Jing Liao

**Affiliations:** 1Department of Medical Statistics and Epidemiology, Sun Yat-sen University, Guangzhou 510080, China; 2School of Nursing and Health, Guangzhou Nanfang College, Guangzhou 510970, China; 3Department of Integrated Profession Management, Guangzhou Center for Disease Control and Prevention, Guangzhou 510440, China; 4Global Health Institute, School of Public Health and Institute of State Governance, Sun Yat-sen University, Guangzhou 510275, China

**Keywords:** COVID-19, diabetes management, couple-based intervention, older adults, community-based participatory research

## Abstract

This study aimed to evaluate the gender-specific effect of a couple-based intervention on the management behaviors and mental well-being of community-dwelling older adults with type 2 diabetes mellitus during the COVID-19 partial lockdown in Guangzhou. Out of 207 participants involved in a prior randomized controlled trial (Trial no. ChiCTR1900027137), 156 (75%) completed the COVID-19 survey. Gendered differences in management behaviors and depressive symptoms between the couple-based intervention group and the patient-only control group were compared by distance to the high-risk areas cross-sectionally and longitudinally using random intercept models. Cross-sectionally, female patients of the intervention group had more positive behavior change scores (*β* = 1.53, *p* = 0.002) and fewer depressive symptoms (*β* = −1.34, *p* = 0.02) than the control group. Over time, female patients lived closer to the high-risk areas (<5 km) and showed decreasing depressive symptoms (*β* = −4.48, *p* = 0.008) in the intervention group vs. the control group. No statistically significant between-group difference was found for males. Females tended to benefit more from the coupled-based intervention than males did, particularly among these closer to the high-risk areas. Chronic disease management can be better sustained with active spousal engagement.

## 1. Introduction

Over the past two years, the COVID-19 pandemic has significantly interrupted healthcare services worldwide [1], particularly in routine diabetes care [2]. China adopted a region-specific, multi-level targeted strategy to implement the dynamic zero-COVID-19 policy [3], whereby community healthcare centers have been at the frontline to identify and manage COVID-19 cases. This strategy has proven to be effective in halting the COVID-19 spread [4] but heavily relies on healthcare and social resources. Community healthcare centers have shifted their focus to nucleic acid testing, vaccination, monitoring, and managing potential outbreaks, whereas other routine primary healthcare services such as chronic disease management may be disrupted or even temporarily stopped [5].

The impact of the COVID-19 lockdown on chronic disease management in community healthcare has been studied. A growing body of studies show that the lockdown was detrimental to type 2 diabetes mellitus (T2DM) patients, leading to decreased physical activity and medical visits [6] and a shortage of medications [7] as well as increased levels of anxiety, stress, and depression [8,9]. However, these studies mostly investigated the impact of COVID-19 lockdown during the widespread lockdown period. A longitudinal study in Ya’an found that physical activity among community-dwelling older adults drastically decreased during the outbreak period and began to rebound seven months later but was still below the pre-pandemic baseline level [10]. In the post-pandemic era, the remaining impact of recurring COVID-19 partial lockdowns on the T2DM management of community-dwelling older adults needs further exploration.

More evidence is also needed on the role of family in supporting older T2DM patients’ daily management during the COVID-19 crisis [11]. Family members, especially spouses, have been found to have great influences on their relative’s or partner’s physical activities, dietary behaviors, and weight control [12,13,14]. These involvements have been found to be gender-specific, such that females benefited more from their partner’s care activities in psychological well-being [15], diet adherence [16], and weight loss [17] than their male counterparts. Few studies so far have investigated the gendered effect of spouse support in chronic disease management during the lockdown period. The extent to which spousal support would buffer the impact of COVID-19 on female and male patients’ management behaviors and mental status remains unknown.

Using the outbreak of COVID-19 in Guangzhou as a natural experiment, this study aimed to explore the gender-specific buffering effect of spousal support on older diabetic patients’ behavioral and psychological outcomes. Building on our prior couple-based T2DM intervention (RCT no. ChiCTR1900027137) [18], we investigated the gender-specific effect of an intervention over the COVID-19 outbreak by comparing the management behaviors and depressive symptoms of older patients in a couple-based intervention group versus a patient-only control group cross-sectionally by distance to the high-risk areas and longitudinally from baseline to 3-month follow-up and the COVID-19 survey.

## 2. Materials and Methods

### 2.1. Study Setting and Design

This study belongs to an umbrella study of a multi-center randomized controlled trial (RCT) on couple-based T2DM management in Guangzhou, China. The study protocol was published previously [18]. Briefly, 207 pairs of T2DM patients and their spouses were recruited from 14 community healthcare centers and were randomized into a couple-based intervention group and a patient-only control group within each center. For the intervention group, both the patients and their spouses received four-week group education targeted at the older couples’ awareness and skills of diabetes care and daily management behaviors (including diabetes and complications, healthy diet, medication, and being active), followed by two-month behavior change boosters via telephone to facilitate intervention adaption, whereas these interventions were only provided to the patients in the control group. The intervention was conducted in each healthcare center between 15 September 2020 and 14 March 2021, followed immediately by a three-month follow-up assessment. A small-scale outbreak of COVID-19 took place in Guangzhou on 21 May 2021. The government timely reacted to the outbreak by adopting a region-specific, multi-level targeted strategy [19], with two streets identified as high-risk areas (i.e., full lockdown management). Our RCT study sites were located at varying distances from these two high-risk areas, ranging from 1.7 to 316.6 km (Figure 1). A COVID-19-specific survey was conducted over the phone to investigate our RCT patients’ behavioral and psychological statuses during the partial lockdown from May to June. The study was approved by the Sun Yat-sen University Institutional Review Board (approval no. 2019–064).

### 2.2. Study Participants

The targeted study participants were the 207 T2DM patients involved in the main RCT study, who were community-dwelling older adults aged 55 years and above, had uncontrolled blood glucose, and cohabited with their spouses. Altogether, 156 patients were successfully reached by phone, rendering a response rate of 75.4%.

### 2.3. Measures

To quantify the degree to which T2DM patients’ management behavior and mental health were affected by the partial lockdown, the current analysis utilized lifestyle-related behavior changes due to the outbreak measured specifically in the COVID-19 survey and two repeated measures of diabetes self-care activities and depressive symptoms across the RCT study period, as illustrated in Figure 2.

#### 2.3.1. Lifestyle-Related Behavior Changes

The impact of the outbreak of COVID-19 on patients’ changes in health behaviors was measured by the lifestyle-related behavior changes questionnaire [20], which has been applied in previous studies [21,22,23]. This questionnaire was developed to assess changes in dietary habits (intake, meal pattern, and snack consumption), physical activity (duration and type), and sleep (duration and quality) during the COVID-19 pandemic. The total score was calculated by adding responses to all items together (range −40 to 36), with a higher score indicating better lifestyle-related behavior changes.

#### 2.3.2. Diabetes Self-Care Activities

We used the Chinese version of the Summary of Diabetes Self-Care Activities (SDSCA) questionnaire to quantify patients’ diabetes management behaviors [24]. The 11-item tool measures the frequency of performing diabetes self-care activities over the last seven days, including healthy diet, exercise, self-monitoring of glucose, foot care, and medication adherence. According to the instructions in the SDSCA manual, the length of seven days is defined, which is just recent enough to not yield inaccurate reports due to recall biases. The sum score ranges from 0 to 77. The more days patients conduct these self-care activities, the higher their scores will be.

#### 2.3.3. Depressive Symptoms

Depressive symptoms were measured by the Chinese version of the 10-item Center for Epidemiological Studies Depression Scale (CESD-10). The CESD-10 scale is a validated and reliable mental health assessment tool for older adults in China [25,26]. Responses to the CESD-10 scale are rated on a Likert scale ranging from 0 to 3: rarely, some days (one–two days), occasionally (three–four days), and most of the time (five–seven days) in the past week. The range of sum scores is 0 to 30, with higher scores indicating a higher level of depressive symptoms [27].

#### 2.3.4. Covariates

The following baseline information was collected regarding participants’ demographic characteristics: age, gender, educational level (i.e., primary school or below, secondary school, and tertiary education or above), and retirement status (i.e., retired/not retired). Distance to the high-risk areas was quantified by geographic latitude and longitude and defined as the smaller one of the two distances between the two high-risk areas and the participant’s home address.

### 2.4. Statistical Analysis

Given the randomized block design of the RCT, we employed multi-level models to separate community-level and patient-level variations. To compare between-intervention group differences in study outcomes gender-specifically, a two-level random intercept model was fitted with cross-sectional measures on behavioral and psychological outcomes as dependent variables and a three-way interaction of group, gender, and distance to the high-risk areas as independent variables. These differences were then examined longitudinally by fitting a three-level random intercept model with the repeated outcome measures over time (i.e., baseline, three-month follow-up assessment, and COVID-19 survey) and a four-way interaction term of group, gender, distance to the high-risk areas, and time as independent variables. Only statistically significant interactions were maintained in the final models and illustrated in graphs. All models were adjusted for baseline age, education, and retirement status. Missing data at the three-month follow-up were imputed by multiple imputations using a Markov Chain Monte Carlo procedure with 20 iterations (19.5% and 15.2% for the intervention and control groups, respectively) [28]. For all analyses, two-sided *p* < 0.05 was considered statistically significant. Statistical analyses were performed using R Studio (Version 2022.2.3.492).

## 3. Results

Table 1 summarizes the basic information of the 156 patients. They were equally recruited from the interventional and control groups of the RCT (response rate: 72.6% vs. 78.2%), with similar between-group sociodemographic characteristics. The participants had a mean age of 66 years, half were male, one-third had tertiary education or above, and the majority of them were retired.

### 3.1. Gender-Specific Differences in Outcomes between Intervention Groups by Distance to the High-Risk Areas for the Cross-Sectional COVID-19 Survey

The three-way interactions of group, gender, and distance to the high-risk areas for all outcomes were not statistically significant (*p*_lifestyle-related behavior changes_ = 0.77, *p*_self-care activities_ = 0.66, *p*_depressive symptoms_ = 0.13) (Appendix A). Figure 3 shows the estimated outcomes by group and gender with distances to the high-risk areas. The closer patients were to the high-risk areas, the lower their lifestyle-related behavior change scores were and the higher their depressive symptom scores were, but these changes with distance were not statistically significant (*p*_s_ > 0.05). Scores on self-care activities were relatively stable over distance.

These associations with distance were similar across the intervention and control groups but varied by gender. For each distance, female patients in the intervention group had higher lifestyle-related behavior change scores (*β* = 1.53, *p* = 0.002) and fewer depressive symptoms (*β* = −1.34, *p* = 0.02) compared to their counterparts in the control group, suggesting a lower impact of the COVID-19 lockdown on the couple-based interventional group. A similar buffering effect of the couple-based intervention on male patients was found in terms of depressive symptom scores only (*β* = −1.34, *p* = 0.02).

### 3.2. Longitudinal Changes of the Gender-Specific Differences in Outcomes between Intervention Groups by Distance to the High-Risk Areas

In the longitudinal analysis of repeat measures, significant interactions of gender, group, time, and distance to the high-risk areas were found for depressive symptoms (*p* = 0.03) but not for self-care activities (*p* = 0.57) (see Appendix A). These results indicate that the intervention effects on the changes in depressive symptoms from the baseline to the COVID-19 survey varied by gender and distance, whereas self-care behaviors were relatively stable.

Figure 4 illustrates these varied changes from the baseline to the COVID-19 survey in depressive symptoms for males and females by three levels of distance to the high-risk areas (i.e., closest distance: <5 km; middle distance: 5~30 km; farthest distance: >30 km). Female patients at the closest distance benefitted from the couple-based intervention in terms of lowering depressive symptoms over time (*β* = −4.48, *p* = 0.008), but those at the middle distance (*β* = −1.87, *p* = 0.28) or the farthest distance (*β* = 3.63, *p* = 0.06) did not. For male patients, none of these between-group differences yielded any statistical significance across the three levels of distance (all *p* > 0.05).

## 4. Discussion

Our study investigated the gender-specific effectiveness of a couple-based intervention over the COVID-19 partial lockdown in Guangzhou. The cross-sectional and longitudinal results revealed that our patients’ diabetes management behaviors and mental well-being were not substantially affected by the partial lockdown, and the effect of the couple-based intervention varied according to gender and distance to high-risk areas. Specifically, females in the intervention group had more positive lifestyle-related behavior changes and reduced depressive symptoms compared to their counterparts of the control group during the lockdown. A statistically significant between-group reduction in depressive symptoms was also shown among female but not male patients who lived closest to the high-risk areas.

We found that the COVID-19 partial lockdown in Guangzhou did not substantially affect the T2DM management and mental health of our study participants. Our finding is different from the results of previous studies conducted during the widespread lockdown period over China in early 2020, which tended to show negative impacts on health outcomes [6,7,10]. This difference may result from the different prevention and control measures, the degree of lockdown restrictions, and also the public’s increased understanding about COVID-19. What is more, in contrast to the hypothesized intervention effect varying by distance to the high-risk areas, the only significant effect of distance on between-group differences was found longitudinally in relation to the depressive symptoms of females living closest to the high-risk areas. It is suspected that most of our participants lived in areas where restriction measures were relatively loose, and thus, their daily activities were only slightly affected. Prior studies on the nationwide lockdown in China also revealed the resilience of older adults against COVID-19. Their findings indicated that older patients with chronic diseases saved more time for self-care activities than social gatherings, such that they would spend more time on physical activities and better adhere to healthy meals [29,30]. Moreover, our findings may reflect the timely control measures implemented in Guangzhou [4]. A series of targeted supporting policies, such as long-term prescription for chronic disease medication [31], designated hospitals for positive COVID-19 patients, and enhanced home-visiting services of primary care providers [32], helped the older patients cope with the shortage of medicine and inaccessibility of medical services caused by COVID-19 outbreaks.

Despite the impact of the COVID-19 partial lockdown on patients being small, we still identified gendered intervention effects on lifestyle-related behavior changes and depressive symptoms. Our results indicated that the couple-based intervention turned out to be more beneficial for female participants than for male participants, whereby more positive changes in lifestyle and fewer depressive symptoms were found among female patients of the intervention group than in those of the control group. A gender-specific effect was reported previously, where female diabetic patients were more likely to lose weight if accompanied by their husbands during the intervention [17]. Furthermore, females are in general more sensitive to the quality of a relationship than their male counterparts [33]. These gendered effects of the couple-based intervention may be further rooted in Chinese Confucian culture and traditional social roles [34]. Chinese men tend to emphasize their masculine traits and be less caring, while women mainly take the caregiver roles [35]. Our previous qualitative study on older Chinese couples with T2DM revealed that wives, regardless of their health status, were more actively involved in or fully responsible for taking care of their husband’s disease management, whereas husbands were less engaged in their wives’ care activities [36]. Older Chinese men highly relied on their partner for daily care, to the extent that these husbands whose spouses had a chronic disease had higher risk of becoming chronically ill over time [37]. Under this circumstance, our couple-based intervention is important as it fosters a sense of responsibility and a great willingness to collaborate as a partner to manage T2DM, particularly for husbands who otherwise are less aware of the needs of their wives. The non-significant changes between male patients of the couple-based vs. patient-only groups may indicate their sufficient spousal support regardless of our intervention. On the contrary, enhanced support from husbands, whether emotional or practical, or just their attendance at education classes may make up the unmet need for spousal involvement for our female participants in the intervention group.

The strength of the current study is its longitudinal design embedded in our RCT and the use of the COVID-19 partial lockdown as a natural experiment. This unique study design enabled us to explore the gender-specific effect of the couple-based intervention on chronic disease management in the post-COVID-19 era. Nevertheless, several limitations should be noted. First, our study sample was composed of community-dwelling older couples where one partner had uncontrolled T2DM. Participants continually involved in the study may be more likely to represent younger old couples with a relatively good relationship. The extent to which our gender-specific findings on the couple-based intervention may be applied to older couples with less satisfactory marital relationships warrants further investigation. Second, we did not measure spousal behavioral and psychosocial changes in the COVID-19 survey and thus cannot quantify the between-group differences in these outcomes among spouses over the COVID-19 partial lockdown. Lastly, self-reported data were collected face-to-face at the study baseline but via telephone for the follow-ups. Our health-related outcomes measured by self-reported questionnaires cannot be exempt from information bias associated with participants’ familiarity with the survey [38] or gendered preference in reporting [39]. Nevertheless, given the randomized controlled study design and gender-specific analyses, any between-group differences by gender would not be affected by potential information bias or the variations in self-reported data collection methods.

## 5. Conclusions

This study investigated the gender-specific effects of a couple-based intervention on T2DM patients’ behavior and mental well-being over the COVID-19 partial lockdown in Guangzhou. Although the partial lockdown did not substantially affect our study participants’ daily activities and care practices, we found a gendered effect showing that females benefited more than males from the couple-based intervention, particularly among those living closest to the high-risk areas. Our findings highlight the importance of husbands’ engagement in chronic disease management and promote an older couple-centered care model to maintain adequate self-management and improve resilience against the impact of pandemics such as COVID-19.

## Figures and Tables

**Figure 1 healthcare-10-02290-f001:**
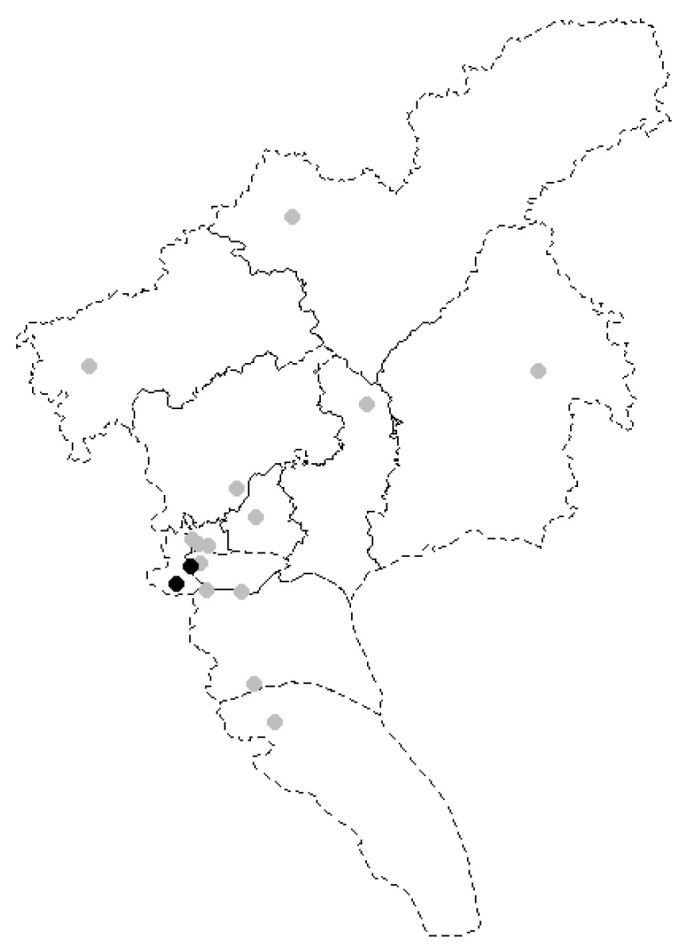
Geographic locations of 14 selected communities for the study (indicated by grey dots) and the two high-risk areas of COVID-19 in Guangzhou (indicated by black dots).

**Figure 2 healthcare-10-02290-f002:**
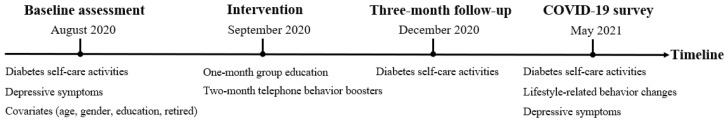
Measures across study period.

**Figure 3 healthcare-10-02290-f003:**
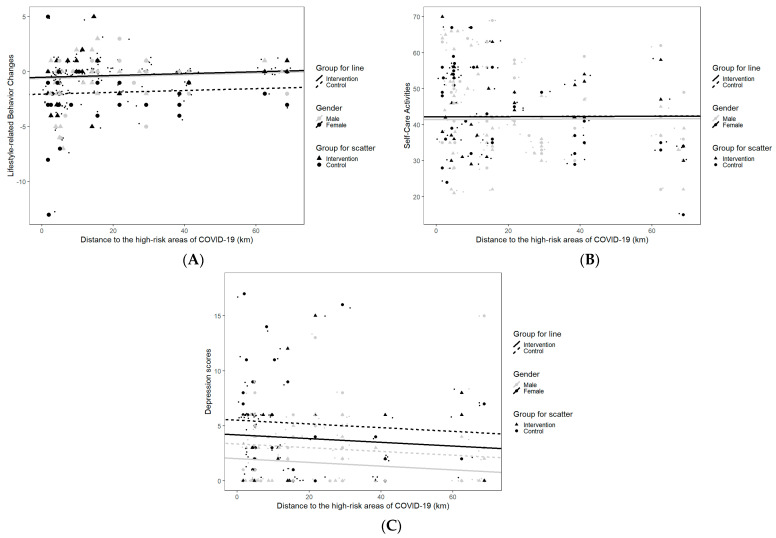
(**A**) Gender-specific differences in lifestyle-related behavior changes between intervention and control groups by distance to the high-risk areas of COVID-19 in Guangzhou. (**B**) Gender-specific differences in self-care activities between intervention and control groups by distance to the high-risk areas of COVID-19 in Guangzhou. (**C**) Gender-specific differences in depressive symptoms between intervention and control groups by distance to the high-risk areas of COVID-19 in Guangzhou.

**Figure 4 healthcare-10-02290-f004:**
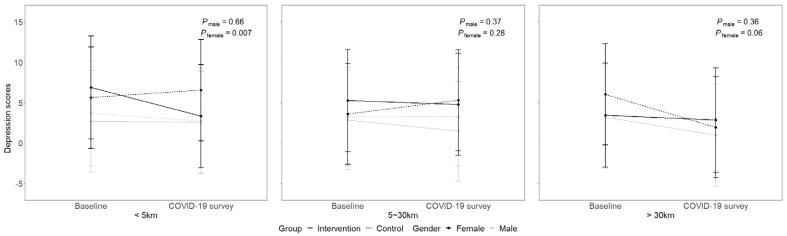
Longitudinal changes in gender-specific differences in depression scores between intervention and control groups by distance to the high-risk areas of COVID-19 in Guangzhou (*p*-values for differences between intervention and control groups for male and female patients).

**Table 1 healthcare-10-02290-t001:** Characteristics of the respondent patients by group.

	Total (n = 156)	Control Group (n = 79)	Intervention Group (n = 77)	*p*-Value for Group Differences
Response rate (%)	75.4	78.2	72.6	0.38
Age M (SD)	66.0 (6.2)	66.1 (6.13)	66.0 (6.36)	0.92
Male (%)	53.8	49.4	58.4	0.26
Education (%)				0.38
Primary school or below	30.8	38.0	23.4	
Secondary school	29.5	21.5	37.7	
Tertiary education or above	39.7	40.5	39.0	
Retired (%)	87.2	84.8	89.6	0.38

## Data Availability

The data presented in this study are available on request from the corresponding author.

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
