# Peer review of "Gender-Specific Effect of Couple-Based Intervention on Behavioral and Psychological Outcomes of Older Adults with Type 2 Diabetes during the COVID-19 Partial Lockdown in Guangzhou, China"

_healthcare, 2022, doi:10.3390/healthcare10112290_

Round 1
Reviewer 1 Report
This study offers interesting insights into the impact of COVID-19 protocols on specific communities and individuals with chronic health conditions. As part of a larger RCT, the study design is strong and the specific methodology for this sub-study well thought out. I found the map helpful in understanding the specific geographic region of the intervention. The 75% response rate to the phone survey is excellent.
My biggest concern relates to the limitations of self-report data. This was not mentioned in the strengths and weakness of the study. Nor, was it mentioned if the researchers took any actions to reduce social desirability bias associated with self-report data. Other studies suggest a gender gap in self-report data with regard to differences in social acceptance desires:
· Rosenman R, Tennekoon V, Hill LG. Measuring bias in self-reported data. Int J Behav Healthc Res. 2011 Oct;2(4):320-332.
· Boerma, T., Hosseinpoor, A.R., Verdes, E. et al. A global assessment of the gender gap in self-reported health with survey data from 59 countries. BMC Public Health 16, 675 (2016).
I recommend adding something in the discussion about how the study design overcame potential bias related to gender differences in self-reported data. Or, noting that self-reported data collection methods coupled with gender differences in reporting may be a confounding variable in the gender differences found in this study.
Reviewer 2 Report
This current study is of importance to chronic diseases management under the context of the COVID-19 pandemic. The authors provided interesting results. But revisions should be made before the manuscript could be accepted for publication.
The major issue:
The research question of this current study was unclear. The authors stated in the Introduction section that they investigated the gender-specific effect of the intervention over the COVID-19 outbreak. But in the Title and the Discussion section, the authors linked patients’ behaviors and mental wellbeing to the partial lockdown strategy. The pandemic per se and the prevention and control strategy were two different concepts.
Other issues:
1. The authors need to briefly present the intervention. Although the authors may have explained the intervention in details in the protocol published, as the authors were conducting an effect evaluation study, they have to give at least an outline of what had been conducted regarding the specific couple-based intervention, e.g., the strategies that were tailored for couples. It was hard to understand the results or make conclusions if no information on the intervention strategies was provided.
2. The authors need to explain explicitly the primary content of each survey and the measures that they used from each survey. It seems that the authors used data from three surveys, i.e., one at baseline, one at the third month after the intervention, and another related to the small-scale COVID-19 outbreak which happened about two-month later after the intervention. No information on when and where the COVID-19 survey was conducted can be found in the manuscript.
3. According to the Statistical analysis section, the authors examined longitudinally the between-intervention group differences in behavioral and psychological outcomes gender-specifically. But as explained in the Measures section, data on various variables were collected in the baseline survey and the three-month follow-up assessment. Collection of data on the depressive symptoms were mentioned only for the baseline survey. What data were collected in the ‘COVID-19 survey’ were unknown. The authors explained that the lifestyle-related behavior changes data were about the impact of COVID-19 on patients’ health behaviors changes. However, it was unclear that whether the behavior changes data were collected in all three surveys, or just in the ‘COVID-19 survey’.
4. The authors have to provide reference(s) for the application of multiple imputations in substituting missing data.
5. Figure 2 was unclear, e.g., the legends of the intervention and control groups cannot be differentiated. The three sub-figures should be presented separately, with units of both horizontal and vertical axes. Figure 3 had the same problems, e.g., male and female participants cannot be differentiated in the sub-figures.
6. The authors did not explain their results explicitly, e.g., in lines 230-231, the authors attributed the gender difference to Chinese Confucius culture and traditional social roles, but just stopped on those two expressions without further elaborations.
7. Regarding lines 231-234, it is hard to understand why females benefited more from the intervention by stating that female partners were fully responsible for taking care of their husbands’ disease management. In that case, the husbands were taken care of and might have better disease management outcomes.
8. In the Discussion section, the authors need to compare this current study finding with previous ones as indicated in the Introduction section – Ya’an study. If similar findings had been achieved already, the authors need to clarify what contributions they made to the literature.
Reviewer 3 Report
Manuscript ID: healthcare-1991541 Dear authors, It is important to investigate the impact of COVID -19 on T2DM on community dwelling older adults/couples. Below are my suggestions:
1. In the title please check the suitability of the term "psychosocial"
2. Figure 2 has three figures, authors should label them individually (A,B,C) for clarity.
3. Check the correct usage of symbols and units
4. Line no: 115 - what is the reason for measuring the frequency of performing diabetes self-care activities for the last seven days? could you share?
5. Depression, especially in older adults, can co-occur with other serious medical illnesses, such as diabetes, cancer, heart disease, and Parkinson’s disease. These conditions are often worse when depression is present, and research suggests that people who have depression and another disease tend to have more severe symptoms of both illnesses. Therefore,
authors can mention some of the depressive symptoms that they collected from the patients in an appropriate place in this manuscript as well.
6. For the most part, the writing seems good but in some places the usage of words and sentences are ambiguous that can be improved
Round 2
Reviewer 2 Report
The authors have made great efforts in revising the manuscript according to the comments. The manuscript can be accepted. Looking forward to reading it in the published version.